# Influence of Fermentation Time on the Phenolic Compounds, Vitamin C, Color and Antioxidant Activity in the Winemaking Process of Blueberry (*Vaccinium corymbosum*) Wine Obtained by Maceration

**DOI:** 10.3390/molecules27227744

**Published:** 2022-11-10

**Authors:** M. Angeles Varo, Maria P. Serratosa, Juan Martín-Gómez, Lourdes Moyano, Julieta Mérida

**Affiliations:** Departamento de Química Agrícola, Edafología y Microbiología, Facultad de Ciencias, Campus de Rabanales, Universidad de Córdoba, Ed. Marie Curie, 14014 Córdoba, Spain

**Keywords:** blueberry, wine, anthocyanins, vitamin C, antioxidant activity

## Abstract

Flavonoid compounds, including anthocyanins and flavan-3-ol derivatives, total tannins, total vitamin C and resveratrol were analyzed by HPLC in blueberry fruits, their skin and pulp, as well as in wines produced from them. Two wines were elaborated, with different times of fermentation. The fruit analysis provided information on the distribution of bioactive compounds in the berries, showing that the skin had the highest concentrations of all compounds. The winemaking process needed a maceration stage to extract these compounds from skins to wine. This maceration process increased the concentration of all compounds and the antioxidant activity values measured by the DPPH assay, but long maceration times decreased the compounds and the antioxidant activity, due to the phenolic compounds that were involved in several reactions, such as polymerization, copigmentation, degradation, formation of pyranoanthocyanins and reactions between anthocyanins and tannins. The sensorial analysis of wines showed that partial fermentation wine had better characteristics than total fermentation wine, although both wines had a high acidity.

## 1. Introduction

In recent decades, interest in the use of berries and their derivatives has increased due to the benefits that these small fruits present to health [1]. Berry fruits have been proven to be a rich source of dietary antioxidants, and their characterization has been of interest for many research groups [2,3,4,5]. In particular, blueberries are one of the most widely consumed fruits in the world, and many studies have demonstrated their high antioxidant activity [6,7]. Some authors have even described that of all the berries, blueberries exhibit the highest antioxidant activity [8,9]. These berries contain many bioactive compounds which are associated with strong antioxidant activity, namely phenolic compounds and vitamin C, which play important roles in human nutrition due to free radical scavenging activities [10]. Most of the research has mainly studied their flavonoid and anthocyanin concentrations, which are the major pigments in [11,12,13,14]. Besides these compounds, blueberries have been studied because of their resveratrol content [11,15,16,17]. The antioxidant properties of these flavonoids include a wide range of biological effects, such as antioxidant, anti-inflammatory, antiallergic, antiulcer, antibiotic and anticarcinogenic [18]. Epidemiological evidence suggests that high consumption of plant-derived flavonoids may provide protection against coronary heart disease [19], stroke [20] and cancer [21].

*Vaccinium* consists of approximately 450 species, and highbush blueberry (*Vaccinium corymbosum*) is one the three major *Vaccinium* domesticated in the twentieth century [22]. Cultivated blueberries were introduced into western Andalusia from North America in the early 1990s, where commercial crops have only been a regular feature since 1995 [23]. In the south of Spain, blueberry cultivation has increased during the past decade, and is likely to continue to do so, but there are some factors that limit the future of blueberries: the lack of chilling hours; a deficit of water (drought); field and postharvest pests and diseases; and reliance on local fresh markets [24].

Windsor blueberries were released from the University of Florida breeding program in 2001. This variety originated as a seedling from the cross FL83-132×‘Sharpblue’, and it was described as a vigorous variety, with stout stems and a semi-spreading growth habit. The berries are very large, and they have good firmness and excellent flavor.

The most abundant fermented fruit beverage is grape wine; however, in recent years, the consumption of other fruit wines has increased, particularly fruits such as blackberries, blackcurrants and blueberries [25,26,27,28]. The production of these wines is similar to the grape wines, and they are a well-known natural source of bioactive compounds as discussed above.

In this sense, the aim of this work was the study of the bioactive compound distribution in blueberry fruits and the relationship of the fermentation time to the anthocyanin and flavonol profiles, as well as the vitamin C concentration and antioxidant activity of wines obtained from Windsor blueberries with a maceration stage including the solid parts of the berries.

## 2. Results and Discussion

### 2.1. Bioactive Compounds of Berries

Blueberries are especially appreciated for their antioxidant activity [4]. Prior, et al. [4] found that this fruit had the highest antioxidant activity of the 42 fruits and vegetables tested. Their antioxidant activity is largely due to the high concentration of phenolic compounds present in these berries, especially flavonoid compounds, including anthocyanins. Numerous studies have examined the polyphenolic composition of these fruits, finding in all of them high concentrations of these types of compounds [13,14].

Table 1 shows the concentrations of anthocyanins (mg/100 g d.m.) measured in the complete berry, as well as in the skin and the pulp. According to the study by Koca and Karadeniz (2009) [29], the fruits of blueberries usually contain 15 kinds of anthocyanins. In this work, 16 compounds were found, four of them being esterified with a glucose molecule, along with five galactosides, four arabinosides and one pentoside derivative, peonidin-3-pentoside. In the same way, two anthocyanidins were identified in blueberry berries (peonidin and malvidin).

As can be seen, anthocyanins were mostly in skin, the main compound being malvidin-3-galactoside (2321 mg/100 g d.m.), which represented more than 30% of total anthocyanins of the skin. The family of galactosides was the majority, accounting for 56.5% of total anthocyanins of the skin (Figure 1). As can be seen in Table 1, five of the main anthocyanins found (delphinidin, cyanidin, petunidin, peonidin and malvidin) came only from this family.

The second major compound in the skin was another derivative of malvidin, malvidin-3-arabinoside (44.8% of arabinosides derivatives and 14.8% of the total anthocyanins). Similarly, the glycosylated derivative was the most abundant malvidin derivative, at a concentration of 332 mg/100 g d.m. The following important compounds were the derivatives of delphinidin and petunidin. Delphinidin-3-galactoside was found in a concentration of 1059 mg/100 g d.m. (14.1% of the total), and delphinidin-3-arabinoside showed a concentration of 728 mg/100 d.m. (9.8% of the total).

The obtained results were in agreement with the data published by Skrede, Wrolstad, and Durst (2000) [30] in highbush blueberries. The authors found that the most abundant compounds were malvidin-3-galactoside (20.2%) and malvidin-3-arabinoside (13.5%). The authors also found delphinidin-3-galactoside (12.3%), delphinidin-3-arabinoside + cyanidine3-galactoside (12.0%), malvidin-3-glucoside (10.6%), petudinin-3-galactoside (9.1%), petunidin-3-glucoside (7.2%), petunidin-3-arabinoside (6.3%) and delphinidin-3-glucoside (5.4%). However, Prior, et al. (2001) [7] found that in lowbush blueberries, malvidin-3-galactoside (14.4%), malvidin-3-glucoside (14.1%), petudinin-3-glucoside (10.7%), delphinidin-3-glucoside (7.8%), and delphinidin-3-galactoside (7.7%) were the main compounds.

In relation to the anthocyanins composition in pulp, only six compounds were quantified: malvidin-3-glucoside, the galactoside derivatives of delphinidin, petunidin and malvidin and the arabinoside derivatives of delphinidin and malvidin (Table 1). As can be seen, the main compounds in pulps were the galactoside and arabinoside derivatives of malvidin, which were the only compounds that exceed 2 mg/100 g d.m.; the concentrations of the other compounds were below 1 mg/100 g d.m.

The galactoside family was the most abundant again, followed by arabinosides and, in this case, only one glucoside was quantified. The anthocyanin concentration in pulp was 1000 times less than in skin (Figure 1). Anthocyanins are compounds present exclusively in skins, and the concentration in pulp is inappreciable. The small concentration in pulp may be a consequence of a transfer in the mechanical pulp–skin separation operation.

The same anthocyanin profile of skin discussed above was found in the berries. The profile showed approximately the same proportions of all the families. The malvidin derivatives were the main compounds, with malvidin-3-galactoside and malvidin-3-arabinoside standing out.

Other phenolic compounds were also measured. Specifically, five flavan-3-ol derivatives were quantified in skin: catechin, epicatechin, epigallocatechin, epigallocatechin gallate and procyanidin B1. Epigallocatechin was not quantified in pulp and berries (Table 1). As can be seen, catechin was the main compound in skin, pulp and berry. These compounds were present in both parts of the fruit, showing the highest concentrations in skin. The berry profile was similar to that found in the pulp, due to blueberry pulp representing more than 90% of fruit weight. Gavrilova, et al. (2011) [12] measured the concentration of phenolic families in four blueberry varieties. They only measured two flavan-3-ol derivatives, catechin and dimer B2, in three of the studied varieties, with their concentrations being much lower than those of other phenolic families.

Other studies show the content of low degree of polymerization flavanols (monomers + dimers + trimers), but do not show the individualized concentrations of the identified compounds ((−)-epicatechin, (+)-catechin, gallocatechins, etc.), only the sum of all of them [31]. The authors measured flavan-3-ol concentrations in 28 fruits, making evident that monomers epicatechin and catechin were prevalent in berries. Specifically, in blueberries with a total concentration of flavan-3-ol derivatives of 44.46 mg/100 g dry weight, the authors indicated that dimers and trimers were the prevailing compounds, a fact that would not be in agreement with the results obtained in this study.

Total tannins were also measured in berries; the highest concentration was found in skin (4.34 g/100 g d.m.), this concentration being three times higher than in pulp. In berries the content was 2.10 g/100 g d.m. (Figure 2a).

Vitamin C is an effective reducing agent with high antioxidant activity [32], but in many studies it was evaluated to contribute only a small amount (up to 10%) to the total antioxidant capacity of the fruits [8,33]. In this regard, total vitamin C concentration was measured in both forms: L-ascorbic acid and dehydroascorbic acid (Figure 2b). As can be seen, the fruit skin showed the highest concentration (77.6 mg/100 g d.m.), with the pulp value being approximately half of that. In both parts of the fruits, approximately 50% of the vitamin C was in the oxidized form. The content of the complete berry was 39 mg/100 g d.m. Namiesnik, et al. (2013) [34] determined total vitamin C in complete lyophilized berries of blueberries, gosseberries and cranberries by CUPRAC assay, and showed the highest values in blueberries (*Vaccinium corymbosum*).

Resveratrol (*trans*-3, 5, 4′-trihydroxystilbene) is a phenolic compound synthetized from cinnamic acid derivatives [35] in response to pathogens and abiotic stress [17]. This compound has been shown to possess a wide range of pharmacological properties, such as chemopreventive activities against cancer [36,37]. The compound presents two isomers, *trans*- and *cis*-resveratrol. *Trans*-resveratrol is the prevailing isomer in grapes (*Vitis vinífera*) and its concentration is higher in skin (50–100 µg/g) [17]. In blueberries studied in this work, *trans*-resveratrol was only quantified in skin and complete berries, with contents of 100 and 44.8 µg/100 g d.m., respectively (Table 1). Other authors have found *t*-resveratrol in different blueberry varieties [15,16]. Rimando, et al. (2004) [16] used lyophilized blueberries and determined concentrations between 7 to 5800 ng/g dry sample, and obtained 1074 ng/g dry sample in *Vaccinum Corymbosum* L. (highbush blueberry).

### 2.2. Bioactive Compounds of Wines

As mentioned above, the most interesting compounds were in fruit skin, so in the elaboration process it was necessary to make a maceration stage with the solid parts, at the same time that the fermentation took place by the yeasts, with the aim to extract them.

Blueberry wines have been widely compared to grape wines. Specifically, Sanchez-Moreno, Cao, Ou and Prior, (2003) [38], compared blueberry wines with red grape wines, and observed that concentrations of total phenolic compounds, total anthocyanins and antioxidant activity were similar.

Red grape wines contain flavonoid compounds, such as flavonols, flavan-3-ol derivatives and anthocyanins [38]. Their different phenolic profile is related with the phenolic composition of fruits and the winemaking conditions, including yeast strain, fermentation temperature and fermentation time. These beverages also contain proanthocyanidins, which are oligomers and polymers of polyhydroxy flavan-3-ol units, and are a significant proportion of the phenolic content [39].

The wines elaborated in this work increased their phenolic compound concentrations with respect to the initial juice (Figure 3). The maceration process with the solid parts favored the compound extraction from skin to juice, so more compounds were quantified. The maceration process has been studied in red wine. The compound extraction from skin increased with the fermentation/maceration time [40,41] and the alcoholic content [40]. The anthocyanins are usually extracted in the first fermentation stage [42].

However, both wines presented different concentrations of phenolic compounds (Figure 4). Wine PF was partially fermented to 6.73% *v*/*v*, and the fermentation was stopped, adding wine alcohol to 12% *v*/*v*, obtaining a sweet wine with 92 g/L of residual sugars. This wine had an anthocyanin content three times higher and flavan-3-ol derivatives content 1.5 times higher than wine TF, which was fermented to consume the sugars by yeasts.

As can be seen, the concentration of the phenolic compound families decreased with the fermentation time. This fact can be due, on the one hand, to an adsorption–desorption balance between the compounds retained in the solid parts of the fruit and those present in the juice solution [43]. On the other hand, anthocyanins and flavan-3-ol derivatives are involved in several reactions, such as pH balance, copigmentation reactions, polymerization, degradation, formation of pyranoanthocyanins and reactions between anthocyanins and tannins [43]. In addition, the anthocyanin loss can be caused by the generation of alcohol during the fermentation process. An increase in the alcohol content results in changes in the joints of the copigmentation product. Therefore, the anthocyanin content decreases due to the adsorption in the solid parts, degradation reactions and structural changes, the last due to the formation of polymeric pigment with tannins.

Table 2 shows the concentration of determined phenolic compounds (mg/L) in the initial juice and in both wines. Galactoside, glucoside and arabinoside derivatives of anthocyanidins malvidin, petunidin, peonidin, cyanidin and delphinidin were determined, in addition to some aglycones and a pentoside derivative. Altogether, 15 anthocyanins were quantified in wines, and only 11 compounds were quantified in initial juice.

Galactoside derivatives were the main family, followed the arabinoside and glucoside derivatives. The compound with the highest concentration in all beverages was malvidin-3-galactoside, which represented between 25 and 30% of the total anthocyanins, followed by malvidin-3-arabinoside and malvidin-3-glucoside. The derivatives of malvidin were the main compounds, being in agreement with the data described above for the fruit. The rest of the anthocyanins were below 1 mg/L.

In relation to flavan-3-ol derivatives, the five compounds described above were quantified, with the main compound being epigallocatechin gallate, followed by catechin. The sum of both compounds represented more than 70% of the family in all beverages.

Tannins influence the sensorial characteristics of wines. They take part in astringency, bitterness, structure, aging capacity and stability of the color of wine. Figure 5a shows a decrease in tannins during the fermentation/maceration of wines. These compounds are present especially in the fruit skin, as mentioned above, and are extracted during fermentation, while at the same time they can participate in several reactions. Tannins can form adducts with anthocyanins, which stabilizes the wine color over time [44,45,46,47] in addition to decreasing the astringency of the tannins [47]. This reaction between tannins and anthocyanins could explain the decrease in both families with the increase in fermentation time.

Total vitamin C also decreased during the winemaking process, although both wines presented similar values (Figure 5b). As can be seen, the reduced form (ascorbic acid) was predominant, due to the reducing medium produced by yeasts during fermentation. Resveratrol could not be quantified in wines (Table 2), probably due to the dilution process that occurred when the sugar solution was added during the elaboration process.

Antioxidant activity is related to phenolic compounds and bioactive compounds [33]. Both wines slightly increased this parameter compared to the initial juice. The latter presented a value of 2.05 mmol TE/L, which increased to 3.54 mmol TE/L for wine PF and obtained a very similar value in wine TF (2.20 mmol TE/L).

The wine with more time of fermentation had fewer bioactive compounds (anthocyanins, flavan-3-ol derivatives and tannins), which was more pronounced in high molecular weight compounds.

In relation to the sensorial characteristics of wines, color is the first attribute to be perceived. In this sense, Table 3 shows the CIELAB coordinates of both wines and the initial juice. As can be seen, the winemaking process decreased the hue angle (h_ab_) to values close to 0º, producing a redness in both wines. However, wine PF increased more in the a* component, so this wine had a more intense red color. Also, a decrease in L* took place, and the wine was darker than the initial juice, with PF being the darkest wine. This change in color could be due to the anthocyanin’s reactions with other compounds, which would be more favored with a longer fermentation time. Some of the new compounds formed present a hypsochromic shift with respect to the starting anthocyanins and a new absorption peak in the 420 nm region, showing orange hues [46].

To finish, both wines were submitted to a tasting panel formed by expert tasters who evaluated the wines in color, aroma and flavor. The most appreciated wine was partial fermentation wine (wine PF), which presented better color and flavor due to the residual sugars. The panel of tasters indicated that both wines had a high acidity.

In conclusion, blueberries were suitable to produce wines with high a concentration of bioactive compounds and antioxidant activity. A large part of the bioactive compounds was found in the skin, so a maceration step with the solid parts of the fruits was necessary in order to extract the largest amount of compounds. However, more time of maceration/fermentation decreased the concentration of all compounds and the antioxidant activity values. The sensorial characteristics of partial fermentation wine (wine PF) were better than those of total fermentation wine (wine TF), although both presented high acidity. Blueberry wines could be a product with a high concentration of healthy bioactive compounds, although it would be necessary optimize the maceration process to extract the highest concentration of these compounds.

## 3. Material and Methods

### 3.1. Material

Highbush blueberry (*Vaccinium corymbosum*) variety Windsor was harvested in the 2018 season in Southern Spain (Huelva, Spain).

### 3.2. Reagents

Hydrochloric acid, metaphosphoric acid, formic acid, acetic acid, methanol, acetonitrile, ethyl acetate, Luff–Schoorl reactive, potassium dichromate and potassium dihydrogen phosphate were purchased from Merck (Madrid, Spain). Wine-based ethanol was purchased from Alcoholes del Sur (Córdoba, Spain). Anthocyanins (malvidin-3-O-galactoside chloride), flavan-3-ol derivative ((+)-catechin, (−)-epicatechin, epigalocatechin, epigalocatechin galate, procyanindin B1), Trolox (vitamin E equivalent) (6-hydroxy-2,5,7,8-tetramethylchroman-2-carboxylic acid), DPPH (2,2,-diphenylpicrylhydrazyl) and DTT (DL-dithiothreitol) were purchased from Sigma-Aldrich Chemical Co. (Madrid, Spain).

### 3.3. Methods of Sample Preparation

#### 3.3.1. Extraction Procedure

The berries were peeled, separating the skin from the pulp. Each fraction, as well as the complete berry, was lyophilized.

In each case, 0.5 g of lyophilized samples were treated with 3 mL of acidified methanol (0.1% HCl), introducing the mixture in ultrasound for 10 min. The supernatant was removed and 3 mL of acidified methanol was added again, performing the procedure three times. All the extract was centrifuged 10 min at 4000 rpm and the supernatant was made up to 10 mL. The extract was filtered with 0.45 μm nylon filter before analysis.

#### 3.3.2. Blueberry Winemaking

Blueberries were mixed and crushed with a sugar solution in a ratio of 1:1 (weight/volume) to obtain a juice of 21 ºBrix. The juice, with the solid parts of the berries, was added with a yeast inoculum of commercial *Saccharomyces cerevisiae* (viniferm CT 007, Agrovin S.A., Spain) in a dose recommended by the manufacturer (0.3 g/L). The mix was divided into four 1 L flasks with 500 mL of juice and immersed in thermostatized water baths at 21 °C. Two of the flasks were completely fermented (wine TF), and in the other two, the fermentation was stopped when the alcoholic content reached between 6 and 7% *v*/*v* by the addition of wine alcohol up to 12% *v*/*v* (wine PF). After fermentation/maceration, the berries were pressed a second time on a vertical press and skin residues removed from the wine. The resulting wines were centrifuged at 3000 rpm, filtered and analyzed in triplicate.

### 3.4. Separation and Extraction of Anthocyanins from Juice/Wine

A volume of 2 mL of juice/wine was passed through a Sep-Pak C18 cartridge, with 900 mg of filling (Long Body Sep-Pak Plus; Waters Associates, Milford, MA, USA) that was previously activated with 5 mL of methanol and washed with aqueous HCl 0.01% (*v*/*v*). The cartridge was washed with 10 mL of 0.01% aqueous HCl and then with 5 mL of ethyl acetate and sequentially anthocyanins were recovered with 5 mL of methanol which was acidified to pH 2 with HCl. The samples were evaporated to dryness using a vacuum centrifuge thermostatted at 35 °C and then dissolved in aqueous HCl 0.01% (*v*/*v*) and 10% methanol acidified to pH 2. Samples were passed through a Nylon filter of 0.45 µm pore size for HPLC analysis.

### 3.5. Identification and Quantification by HPLC-DAD of Anthocyanins

A volume of 20 μL of the sample were injected into a P4000 HPLC instrument from Spectra-Physics (San Jose, CA, USA) and were identified by comparing their retention times with those for standards, recording UV spectra on a Spectra-Physics UV6000LP diode array spectrophotometer and calculating the UV absorbance ratios for samples and standards simultaneously co-injected one at a time. Identifications were confirmed by HPLC–ESI–MS on an AQA quadrupole mass spectrometer from Thermo. The instrument was operated in both the negative and positive ion modes. The ion spray voltage was −4 kV and the orifice voltage was −60 V. Mass data were acquired in scan mode (by scanning the m/z range 150–1066 at 1.2 intervals per second) and multiple ion mode (by using mass ranges around specific m/z values).

Analyses were carried out on a LiChrospher 100 RP-18 column (250 mm × 4.6 mm, 5 µm), using 10% aqueous formic acid in HPLC-grade water (solvent A) and 10% formic acid, 45% acetonitrile and 45% HPLC-grade water (solvent B), as a mobile phase, at a flow rate of 1 mL/min. Anthocyanins were registered at 520 nm, by gradient elution from 15% to 30% B in 17 min, gradient elution up to 73% B in 28 min, gradient elution up to 100% B in 3 min and isocratic elution for 3 min. All anthocyanins were quantified as malvidin-3-O-galactoside.

### 3.6. Identification and Quantification by HPLC-F of Flavan-3-ol Derivatives and Resveratrol

In the case of identification and quantification of flavan-3ol derivatives, the samples were diluted 10 times in ultrapure water for extract and 20 times for wines. Resveratrol was only found in extract. For *t*-resveratrol, 5 mL of extract were evaporated to dryness and redissolved in 1 mL of acetonitrile/5% acetic acid solution (9:91). The identification and quantification were carried out in a HPLC (Thermo Spectra Physic Series P100) with fluorescence detector (Perkin Elmer Series 200a), on a LiChrospher 100 RP-18 column (250 mm × 4.6 mm, 5 µm), using acetonitrile (solvent A) and 5% acetic acid (solvent B), as a mobile phase, at a flow rate of 1.4 mL/min and the following elution program: gradient 9 to 25% of solvent A in 22 min followed by an increase to 100% of solvent A in 8 min and isocratic elution for 2 min. For flavan-3-ol derivatives analysis λ_exc_ = 280 nm and λ_em_ = 320 nm were used, and for resveratrol analysis λ_exc_ = 324 nm and λ_em_ = 370 nm were used.

### 3.7. Vitamin C

In the case of blueberry wine, vitamin C was identified and quantified by the Sdiri Navarro, Monterde, Benabda and Salvador (2012) method [48]. A quantity of 0.7 mL of 4.5% metaphosphoric acid was added to 0.7 mL of blueberry wine. For pulp, skin and whole blueberry extracts, 2 mL of 4.5% metaphosphoric acid was added to 0.1 g of lyophilized sample. The mixtures were introduced into ultrasonic bath for 5 min. The supernatant was centrifuged at 4000 rpm 5 min. A quantity of 1 mL of the supernatant was added to 0.2 mL of DTT solution and the sample was kept dark for 2 h in order to reduce the dehydroascorbic acid to L-ascorbic acid. After complete conversion, the sample was filtered with 0.45µm nylon filter pore size. Ascorbic acid quantification was performed on a HPLC chromatograph (Thermo Spectra Physics Series P100) coupled to a UV detector (Thermo Finnigan Spectra System UV2000), using a LiChrospher 100 RP-18 column (250 mm × 4.6 mm, 5 µm). KH_2_PO_4_ (0.2 M at pH = 2.3–2.4) was used as mobile phase with a flow of 1.0 mL/min for 15 min at λ = 243 nm and an injection volume of 20 µL.

### 3.8. Reducing Sugars

This parameter was determined according to the EEC official methods as described in Regulation 2676/1990. The method used is the Luff–Schoorl method, which is based on the reduction by them of the Cu^2+^ contained in an alkaline solution. After the addition of KI in acid medium, the I_2_ formed by the action of Cu^2+^ is titrated with a 0.1 N Na_2_S_2_O_3_ solution.

### 3.9. Spectrophotometric Determinations

#### 3.9.1. Color

Spectrophotometric measurements were made on a PerkinElmer (Waltham, MA, USA) Lambda 25 spectrophotometer, using quartz cells of 1 mm light path. Samples were previously passed through Millipore (Billerica, MA, USA) HA filters of 0.45 µm pore size. All measurements were corrected for a path length of 1 cm.

CIELAB parameters were carried out following CIE recommendations [49] and using the visible spectrum obtained from 380 to 780 nm. In this work, the following CIELAB uniform space colorimetric parameters have been considered: rectangular coordinates L* (black–white component, lightness), a* and b* (chromatic coordinates representing red–green and yellow–blue axes, respectively) and the cylindrical coordinate h_ab_ (hue angle). These parameters were measured using the CIE 1964 Standard Observer (10° visual field) and the CIE standard illuminant D65 as references.

#### 3.9.2. Total Tannins

The total tannins were determined by the absorbance measurement at 550 nm on a PerkinElmer (Waltham, MA, USA) Lambda 25 spectrophotometer, after acid hydrolysis with HCl of the samples diluted 1:50 and a blank. The resulted absorbance (A_sample_-A_blank_) was multiplied by a factor of 19.33, in order to calculate the total tannin concentration, in g/L in wines [50]. In berry analysis, the values were expressed in g per 100 g of dry matter (g/100 g d.m.).

#### 3.9.3. Antioxidant Activity

Antioxidant activity was analyzed through the DPPH assay according to Alen-Ruiz, García-Falcon, Pérez-Lamela, Martínez-Carballo and Simal-Gandara (2009) [51], with some modifications. A 45 mg/L solution of DPPH in methanol was prepared on a daily basis and stored in the dark. The extract obtained previously was diluted 1:5 with water. An 80 mg/L solution of Trolox, a vitamin E analogue, was used as a standard. The analytical procedure was as follows: a 200 μL aliquot of diluted sample was placed in a cell and 3 mL of a 45 mg/L solution of DPPH in methanol was then added. A blank (200 μL diluted sample + 3 mL methanol), a control sample (200 μL of 12% ethanol in water + 3 mL of DPPH solution) and a Trolox standard (200 μL of Trolox solution + 3 mL of DPPH solution) were also prepared in parallel. Following vigorous stirring, the absorbances at 517 nm of the control sample and the blank were measured on a PerkinElmer (Waltham, MA) Lambda 25 spectrophotometer. The sample and the Trolox standard were measured under identical conditions after 120 min of incubation at room temperature. The results were expressed in millimoles of Trolox per 100 g of dry matter (mmol TE/100 g d.m.) and in millimoles of Trolox per liter (mmol TE/L) in berry and wine analyses, respectively.

#### 3.9.4. Ethanol Content of Wines

This was determined according to Crowell and Ough, (1979) [52]. To this end, ethanol in the sample was collected by steam and then reacted with acid potassium dichromate. The reaction was spectrophotometrically monitored via the absorbance at 600 nm against a blank on a PerkinElmer (Waltham, MA, USA) Lambda 25 spectrophotometer.

### 3.10. Statistical Procedures

The results for all samples were subjected to one-factor analysis of variance test (ANOVA) with a 95% confidence level using the Statgraphics Centurion from Statistical Graphics Corp. It establishes homogeneous groups and allows to check if there are significant differences between groups. The analysis was made with the triplicate of measurements.

## Figures and Tables

**Figure 1 molecules-27-07744-f001:**
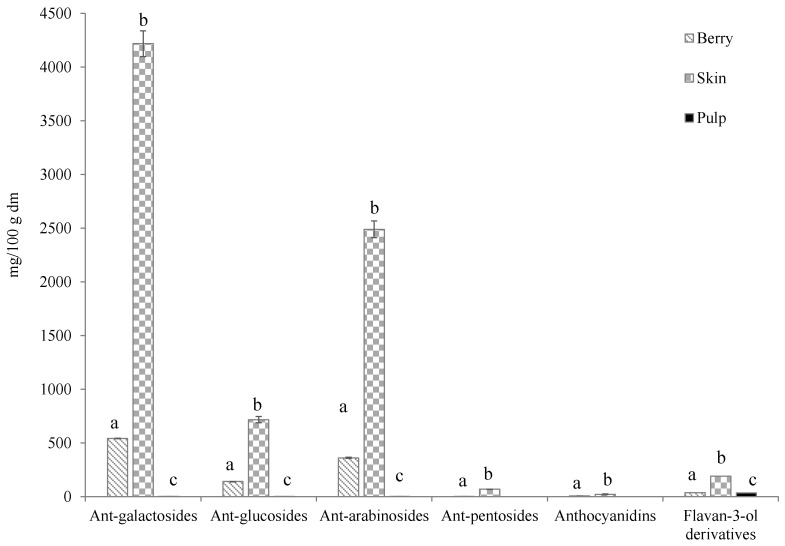
Concentrations (mg/100 g d.m.) of 3-monogalactosides, 3-monoglucosides, 3-arabinosides and 3-pentosides derivatives of anthocyanins and anthocyanidins and flavan-3-ol derivatives in berries, skin and pulp of Windsor blueberry (means and standard deviations). Values with different superscript letters are significantly different, *p* = 0.01.

**Figure 2 molecules-27-07744-f002:**
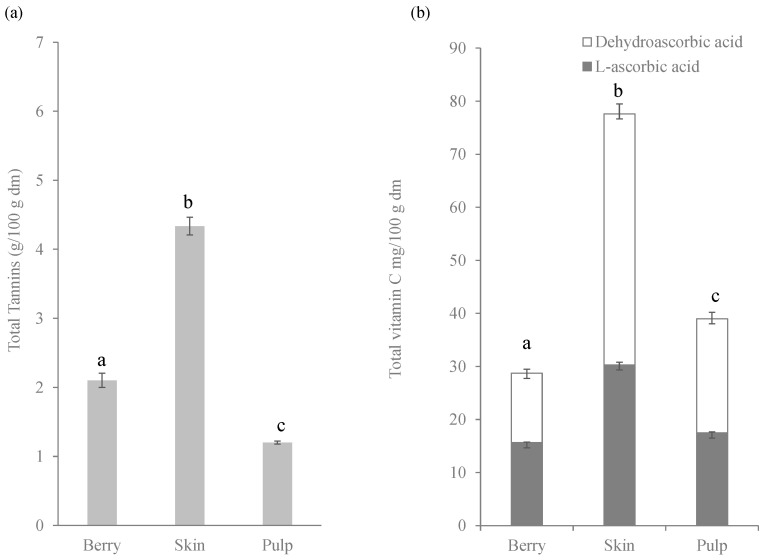
Concentration (g/100 g d.m.) of total tannins (**a**) and (mg/100 g d.m.) of total vitamin C (**b**) in berries, skin and pulp of Windsor blueberry (means and standard deviations). Values with different superscript letters are significantly different, *p* = 0.01.

**Figure 3 molecules-27-07744-f003:**
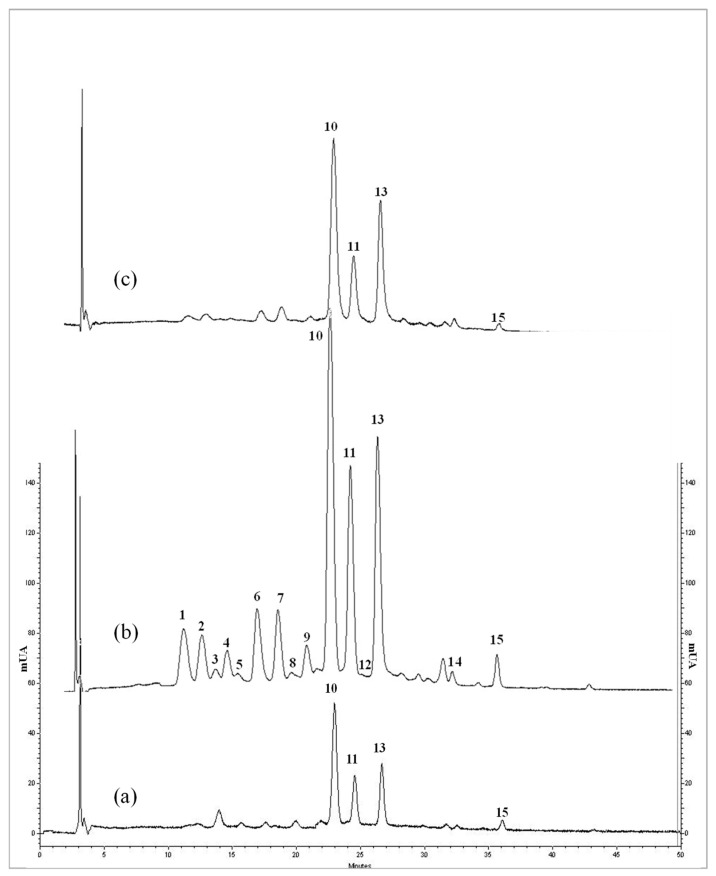
Chromatograms of the anthocyanins fraction corresponding to initial juice (**a**), partial fermentation wine (wine PF) (**b**) and total fermentation wine (wine TF) (**c**). Anthocyanins: 1. Dp-3-gal: Delphinidin-3-galactoside; 2. Dp-3-glc: Delphinidin-3-glucoside; 3. Cn-3-gal: Cyanidin-3-galactoside; 4. Dp-3-arb: Delphinidin-3-arabinoside; 5. Cn-3-glc: Cyanidin-3-glucoside; 6. Pt-3-gal: Petunidin-3-galactoside; 7. Cn-3-arb: Cyanidin-3-arabinoside; 8. Pn-3-gal: Peonidin-3-galactoside; 9. Pt-3-arb: Petunidin-3-arabinoside; 10. Mv-3-gal: Malvidin-3-galactoside; 11. Mv-3-glc: Malvidin-3-glucoside; 12. Pn-3-pentoside: Peonidin-3-pentoside; 13. Mv-3-arb: Malvidin-3-arabinoside; 14. Pn: peonidin 15. Mv: malvidin.

**Figure 4 molecules-27-07744-f004:**
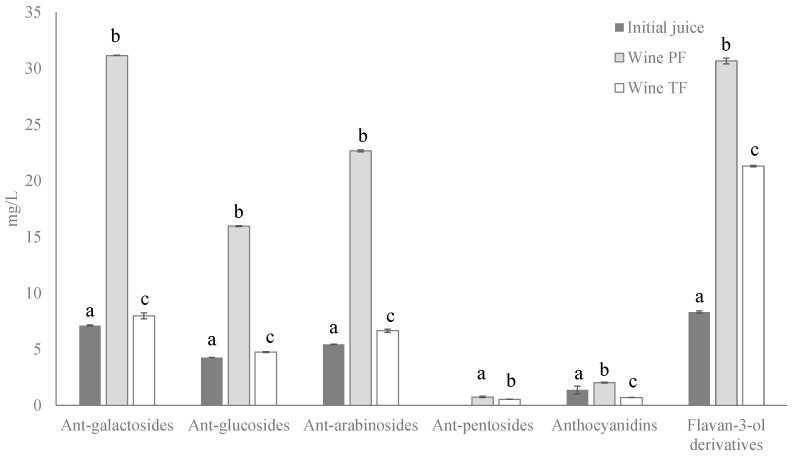
Concentration (mg/L) of 3-monogalactosides, 3-monoglucosides, 3-arabinosides and 3-pentosides derivatives of anthocyanins and anthocyanidins and flavan-3-ol derivatives in initial juice and wines from Windsor blueberry (means and standard deviations). Values with different superscript letters are significantly different, *p* = 0.01.

**Figure 5 molecules-27-07744-f005:**
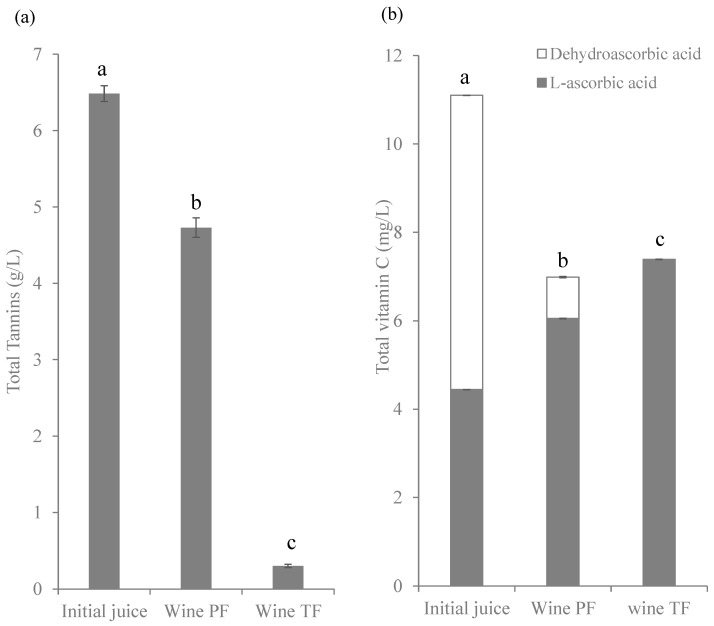
Concentration (g/L) of total tannins (**a**) and (mg/L) total vitamin C (**b**) in initial juice and wines from Windsor blueberry. Values with different superscript letters are significantly different, *p* = 0.01.

**Table 1 molecules-27-07744-t001:** Flavonoid contents (mg/100 g d.m.) and resveratrol in berry, skin and pulp of Windsor blueberry (means, standard deviations and homogenous groups).

	Berry	Skin	Pulp
Dp-3-gal	187 ± 2.90 ^a^	1059 ± 14.0 ^b^	0.448 ± 0.148 ^c^
Cn-3-gal	20.0 ± 3.60 ^a^	162 ± 16.1 ^b^	n.d.
Pt-3-gal	120 ± 0.702 ^a^	604 ± 24 ^b^	0.532 ± 0.164 ^c^
Pn-3-gal	7.10 ± 0.430 ^a^	73.6 ± 4.8 ^b^	n.d.
Mv-3-gal	209 ± 1.50 ^a^	2321 ± 75.7 ^b^	2.91 ± 0.012 ^c^
Dp-3-glc	47.9 ± 0.503 ^a^	178 ± 9.00 ^b^	n.d.
Cn-3-glc	4.3 ± 0.031 ^a^	22.4 ± 3.82 ^b^	n.d.
Pt-3-glc	n.d.	186 ± 4.20 ^b^	n.d.
Mv-3-glc	88.5 ± 1.70 ^a^	332 ± 14.0 ^b^	0.288 ± 0.128 ^c^
Dp-3-arb	124 ± 1.27 ^a^	728 ± 8.00 ^b^	0.54 ± 0.101 ^c^
Cn-3-arb	50.5 ± 0.115 ^a^	188 ± 20.0 ^b^	n.d.
Pt-3-arb	66.8 ± 1.60 ^a^	460 ± 28.0 ^b^	n.d.
Mv-3-arb	121 ± 2.20 ^a^	1115 ± 30.0 ^b^	2.57 ± 0.170 ^c^
Pn-3-pentoside	1.31 ± 0.192 ^a^	69.5 ± 0.231 ^b^	n.d.
Pn	3.82 ± 0.423 ^a^	n.d.	n.d.
Mv	3.31 ± 0.110 ^a^	21.8 ± 3.53 ^b^	n.d.
Total anthocyanins	1054 ± 8.80 ^a^	7519 ± 230 ^b^	7.30 ± 0.381 ^c^
Catechin	25.4 ± 0.234 ^a^	98.1 ± 3.64 ^b^	23.6 ± 0.110
Epicatechin	4.86 ± 0.038 ^a^	12.9 ± 0.559 ^b^	4.11 ± 0.026
Epigallocatechin	n.d.	13.9 ± 1.09 ^b^	n.d.
Epigallocatechin-gallate	5.84 ± 0.074 ^a^	57.3 ± 3.02 ^b^	6.41 ± 0.016
Procyanidin B1	1.69 ± 0.006 ^a^	10.5 ± 0.131 ^b^	1.04 ± 0.012
Total flavan-3-ol derivatives	37.8 ± 0.267 ^a^	193 ± 2.13 ^b^	35.1 ± 0.061 ^c^
Resveratrol	44.8 ± 0.623 ^a^	100 ± 0.680 ^b^	n.d.

Dp-3-gal: Delphinidin-3-galactoside; Cn-3-gal: Cyanidin-3-galactoside; Pt-3-gal: Petunidin-3-galactoside; Pn-3-gal: Peonidin-3-galactoside; Mv-3-gal: Malvidin-3-galactoside; Dp-3-glc: Delphinidin-3-glucoside; Cn-3-glc: Cyanidin-3-glucoside; Pt-3-glc: Petunidin-3-glucoside; Pn-3-pentoside: Peonidin-3-pentoside; Mv-3-glc: Malvidin-3-glucoside; Dp-3-arb: Delphinidin-3-arabinoside; Cn-3-arb: Cyanidin-3-arabinoside; Pt-3-arb: Petunidin-3-arabinoside; Pn-3-arb: Peonidin-3-arabinoside; Mv-3-arb: Malvidin-3-arabinoside; Pn: peonidin; Mv: malvidin; n.d.: not detected. Values in the same row with different superscript letters are significantly different, *p* = 0.01. Structures of Catechin, Epicatechin, Epigallocatechin, Epigallocatechin-gallate, Procyanidin B1 and Resveratrol shown in Appendix A.

**Table 2 molecules-27-07744-t002:** Flavonoid contents (mg/L) and resveratrol in initial juice and wines from Windsor blueberry (means, standard deviations and homogenous groups).

	Initial Juice	Wine PF	Wine TF
Dp-3-gal	0.642 ± 0.004 ^a^	4.02 ± 0.047 ^b^	0.472 ± 0.027 ^c^
Cn-3-gal	n.d.	1.07 ± 0.084 ^a^	0.422 ± 0.006 ^b^
Pt-3-gal	0.650 ± 0.005 ^a^	4.83 ± 0.041 ^b^	0.822 ± 0.006
Pn-3-gal	n.d.	1.18 ± 0.041 ^a^	0.465 ± 0.031 ^b^
Mv-3-gal	5.83 ± 0.047 ^a^	20.0 ± 0.124 ^b^	5.80 ± 0.261 ^a^
Dp-3-glc	1.31 ± 0.002 ^a^	3.38 ± 0.011 ^b^	0.663 ± 0.004
Cn-3-glc	n.d.	0.938 ± 0.060 ^a^	0.468 ± 0.034 ^b^
Mv-3-glc	2.95 ± 0.005 ^a^	11.6 ± 0.035 ^b^	3.62 ± 0.076 ^c^
Dp-3-arb	0.595 ± 0.002 ^a^	2.33 ± 0.105 ^b^	0.498 ± 0.022 ^a^
Cn-3-arb	0.801 ± 0.005 ^a^	4.34 ± 0.012 ^b^	1.23 ± 0.016 ^c^
Pt-3-arb	0.825 ± 0.010 ^a^	2.63 ± 0.073 ^b^	0.676 ± 0.030 ^c^
Mv-3-arb	3.23 ± 0.016 ^a^	13.4 ± 0.128 ^b^	4.25 ± 0.181 ^c^
Pn-3-pent	n.d.	0.759 ± 0.061 ^a^	0.546 ± 0.009 ^b^
Pn	0.734 ± 0.343 ^a^	0.549 ± 0.017 ^a^	n.d.
Mv	0.646 ± 0.004 ^a^	1.48 ± 0.022 ^b^	0.710 ± 0.013 ^c^
Total Anthocyanins	18.21 ± 0.399 ^a^	72.5 ± 0.109 ^b^	21.0 ± 0.437 ^c^
Catechin	2.74 ± 0.036 ^a^	10.5 ± 0.260 ^b^	6.40 ± 0.205 ^c^
Epicatechin	0.364 ± 0.019 ^a^	1.54 ± 0.040 ^b^	1.40 ± 0.233 ^b^
Epigallocatechin	0.917 ± 0.084 ^a^	3.04 ± 0.100 ^b^	3.40 ± 0.196 ^c^
Epigallocatechin-gallate	3.15 ± 0.008 ^a^	13.4 ± 0.222 ^b^	8.85 ± 0.026 ^c^
Procyanidin B1	1.15 ± 0.070 ^a^	2.15 ± 0.133 ^b^	1.25 ± 0.185 ^a^
Total Flavan-3-ol derivatives	8.32 ± 0.102 ^a^	30.7 ± 0.258 ^b^	21.3 ± 0.067 ^c^
Resveratrol	n.d	n.d.	n.d.

Dp-3-gal: Delphinidin-3-galactoside; Cn-3-gal: Cyanidin-3-galactoside; Pt-3-gal: Petunidin-3-galactoside; Pn-3-gal: Peonidin-3-galactoside; Mv-3-gal: Malvidin-3-galactoside; Dp-3-glc: Delphinidin-3-glucoside; Cn-3-glc: Cyanidin-3-glucoside; Pt-3-glc: Petunidin-3-glucoside; Pn-3-pentoside: Peonidin-3-pentoside; Mv-3-glc: Malvidin-3-glucoside; Mv-3-glc: Malvidin-3-glucoside; Dp-3-arb: Delphinidin-3-arabinoside; Cn-3-arb: Cyanidin-3-arabinoside; Pt-3-arb: Petunidin-3-arabinoside; Pn-3-arb: Peonidin-3-arabinoside; Mv-3-arb: Malvidin-3-arabinoside; Pn: peonidin; Mv: malvidin; n.d.: not detected. Values in the same row with different superscript letters are significantly different, *p* = 0.01.

**Table 3 molecules-27-07744-t003:** CIELAB coordinates of initial juice and wines from Windsor blueberry.

	L*	a*	b*	h_ab_
Initial Juice	68.5 ± 0.105 ^a^	68.4 ± 1.01 ^a^	41.2 ± 0.093 ^a^	31.1 ± 0.418 ^a^
Wine PF	62.1 ± 0.123 ^b^	83.4 ± 0.500 ^b^	38.3 ± 0.094 ^b^	24.7 ± 0.082 ^b^
Wine TF	67.7 ± 0.008 ^c^	71.1 ± 0.003 ^c^	30.9 ± 0.090 ^c^	23.4 ± 0.209 ^c^

Values in the same row with different superscript letters are significantly different, *p* = 0.01.

## Data Availability

Not applicable.

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
