# Peer review of "Influence of Fermentation Time on the Phenolic Compounds, Vitamin C, Color and Antioxidant Activity in the Winemaking Process of Blueberry (*Vaccinium corymbosum*) Wine Obtained by Maceration"

_molecules, 2022, doi:10.3390/molecules27227744_

Round 1
Reviewer 1 Report
The manuscript is interesting for a large audience and is very useful for all wine producers. Before publication, some corrections should be made.
P.4, line 124: what flavanols of low degree of polymerization? Please, specify.
P.8, lines 206-208: Please, add the references.
Figure 6 is not clear. The presentation of data in the form of a table could be better.
Section 3.5: Please, briefly describe the determination of reducing sugars.
Section 2.1 (Table 1): Structural formulas could be provided for flavonoid compounds or for each family. It could help the readers and improve their understanding of the manuscript.
Author Response
P.4, line 124: what flavanols of low degree of polymerization? Please, specify.
- Some compounds have been included in the text
P.8, lines 206-208: Please, add the references.
- A reference has been included
Figure 6 is not clear. The presentation of data in the form of a table could be better.
- Figure 6 has been deleted and the data include in Table 3. The text has been corrected
Section 3.5: Please, briefly describe the determination of reducing sugars.
- The method has been described.
Section 2.1 (Table 1): Structural formulas could be provided for flavonoid compounds or for each family. It could help the readers and improve their understanding of the manuscript.
- A new figure with the structural formulas has been added as supplementary material
Reviewer 2 Report
In this study, anthocyanins, flavan-3-ol derivatives, total tannins, vitamin C, and resveratrol were measured in blueberry fruits, skin, pulp, and two wines. The authors found that the qualities of partial fermentation wine were superior to those of entire fermentation wine, which would be a future reference for the wine fermentation industry.
For the result section, I would suggest the authors to add the sample size in the caption of figures. Clarify error bar represent how many SD. Additionally, for Figure 3, the a,b,c should also be labeled in the figures instead of only mentioning in the caption.
Furthermore, I would suggest the authors to do the statistical analysis between the groups they were comparison, instead of just simply claimed that what value is high or what is similar. ANOVA and post-hoc pairwise comparison should be needed for all the results the authors showed in Figure 1, 2, 4, and 5.
For the introduction section, the authors only cited one recent work published in 2021. I would suggest them to do more literature review and get a better view of the recent research in the same area, which can help them to illustrate the value of their work better.
Author Response
In this study, anthocyanins, flavan-3-ol derivatives, total tannins, vitamin C, and resveratrol were measured in blueberry fruits, skin, pulp, and two wines. The authors found that the qualities of partial fermentation wine were superior to those of entire fermentation wine, which would be a future reference for the wine fermentation industry.
For the result section, I would suggest the authors to add the sample size in the caption of figures. Clarify error bar represent how many SD. Additionally, for Figure 3, the a,b,c should also be labeled in the figures instead of only mentioning in the caption.
- The figures have been corrected, the label a, b, c, has been added, there was a mistake in the edition of the figure
Furthermore, I would suggest the authors to do the statistical analysis between the groups they were comparison, instead of just simply claimed that what value is high or what is similar. ANOVA and post-hoc pairwise comparison should be needed for all the results the authors showed in Figure 1, 2, 4, and 5.
- The ANOVA analysis has been added and the homogeneous group too, to identify the significant differences
For the introduction section, the authors only cited one recent work published in 2021. I would suggest them to do more literature review and get a better view of the recent research in the same area, which can help them to illustrate the value of their work better.
- There are not many references on blueberry wines, there are references on the fruit, but there are fewer references on derived products.
Reviewer 3 Report
This research article investigated the bioactive compounds in blueberry fruits and wines. It reported the anthocyanins, flavan-3-ol derivatives, total tannins, total vitamin C, and resveratrol in fruits (berry, skin, and pulp) and wines (total and partially fermented). The antioxidant activity of wines was also studied using DPPH. The results showed that in fruit, the skin had the highest bioactive compound content, while in wine, the maceration time influenced both compound content and antioxidant activity.
This article is interesting since it reports the effect of long maceration on wine quality (chemistry, sensorial) and could be used to improve this process.
However, I still have some remarks:
While studying the bioactive compounds of berries, the authors have found that the skin had higher content than berry and pulp. However, since the berry is in fact skin+pulp (as explained in sample preparation), it should have higher content (or almost equal) to skin+pulp content. How do the authors explain those results? and the significant difference between berry and skin content.
Also, the article needs a statistical analysis. It will better compare the compound content of different fruit parts and of juice and wines.
Also, it would be interesting to add a comparison between fruit content and juice-wine content.
Specific remarks:
Abstract: please add the techniques used (DPPH ...).
Line 56: explain 'they'.
Line 61: 'higher' than what?
Lines 72-73: not 30% but 14.8% (1115 in a total of 7519).
Lines 75-76: this idea is not clear. Please explain.
Figure 1: the pulp content is not clear.
Lines 81-88: it's not clear that you are always talking about the skin. Add 'skin' in each paragraph or merge all paragraphs.
Line 117: 'catechin was the main compound in skin and pulp' and berry too
Lines 148-158: add 'table 1'.
Line 154: '100 y 44.8 μg/100 g d.m., respectively'. Correct y: and
Figure 3: you need to add a, b, and c on the graph (like figure 2 a and b)
Lines 185-192: It is not clear that these lines are with fig 3. Also, it is better to put them in order (1, 2, 3 ...). Please revise them.
Lines 206-213: these ideas need a reference.
Line 218: peonidin-3-glucoside is absent in table 2. so it wasn't quantified in juice too?
Lines 243-244: add 'table 2'.
3.2. Reagents: please revise. Some were not added: ethyl acetate, vitamin E, ethanol, acid potassium dichromate. Others were not used: phosphoric acid.
Sections 3.3.4. and 3.3.5: it's better to separate them from section 3.3 : Methods of sample preparation. These sections explain the analysis and not the sample preparation.
Section 3.5. Reducing sugars: please add details. Also, the results of this section were not reported.
Line 388: what kind of acid was used for hydrolysis.
Section 3.6.4. Ethanol content of wines: please add the results of this section.
Author Response
This research article investigated the bioactive compounds in blueberry fruits and wines. It reported the anthocyanins, flavan-3-ol derivatives, total tannins, total vitamin C, and resveratrol in fruits (berry, skin, and pulp) and wines (total and partially fermented). The antioxidant activity of wines was also studied using DPPH. The results showed that in fruit, the skin had the highest bioactive compound content, while in wine, the maceration time influenced both compound content and antioxidant activity.
This article is interesting since it reports the effect of long maceration on wine quality (chemistry, sensorial) and could be used to improve this process.
However, I still have some remarks:
While studying the bioactive compounds of berries, the authors have found that the skin had higher content than berry and pulp. However, since the berry is in fact skin+pulp (as explained in sample preparation), it should have higher content (or almost equal) to skin+pulp content. How do the authors explain those results? and the significant difference between berry and skin content.
- The data refer to dry matter of each fraction, i.e., the skin was removed and freeze-dried, expressing its content referring only to the mass of skin. The same was done with the pulp, expressing the contents of phenolic compounds only referred to the mass of pulp. When the whole berry was analyzed, it was completely freeze-dried and pulverized, so that the phenolic compound data for the skin were diluted to the whole mass of the whole berry.
Also, the article needs a statistical analysis. It will better compare the compound content of different fruit parts and of juice and wines.
- The ANOVA analysis has been added and the homogeneous group too, to identify the significant differences. The text has been corrected
Also, it would be interesting to add a comparison between fruit content and juice-wine content.
Specific remarks:
Abstract: please add the techniques used (DPPH ...).
Line 56: explain 'they'.
- The sentence has been rewritten
Line 61: 'higher' than what?
- The sentence has been rewritten
Lines 72-73: not 30% but 14.8% (1115 in a total of 7519).
- The sentence has been rewritten for better understanding.
Lines 75-76: this idea is not clear. Please explain.
- The sentence has been rewritten for better understanding.
Figure 1: the pulp content is not clear.
- The figure has been modified to try to make it look better. The pulp contents were very low, even some family was not quantified in it. This can be seen in Table 1
Lines 81-88: it's not clear that you are always talking about the skin. Add 'skin' in each paragraph or merge all paragraphs.
- The paragraphs have been rewritten for better understanding.
Line 117: 'catechin was the main compound in skin and pulp' and berry too
- The sentence has been rewritten for better understanding.
Lines 148-158: add 'table 1'.
- Table 1 has been added
Line 154: '100 y 44.8 μg/100 g d.m., respectively'. Correct y: and
- The sentence has been corrected
Figure 3: you need to add a, b, and c on the graph (like figure 2 a and b)
- The figures have been corrected, the label a, b, c, has been added, there was a mistake in the edition of the figure
Lines 185-192: It is not clear that these lines are with fig 3. Also, it is better to put them in order (1, 2, 3 ...). Please revise them.
- The anthocyanins have been revised and put them in order
Lines 206-213: these ideas need a reference.
- A reference has been added
Line 218: peonidin-3-glucoside is absent in table 2. so it wasn't quantified in juice too?
- This compound wasn’t quantified, the sentence has been corrected
Lines 243-244: add 'table 2'.
- Table 2 has been added
3.2. Reagents: please revise. Some were not added: ethyl acetate, vitamin E, ethanol, acid potassium dichromate. Others were not used: phosphoric acid.
- The reagents have been revised and completed.
Sections 3.3.4. and 3.3.5: it's better to separate them from section 3.3 : Methods of sample preparation. These sections explain the analysis and not the sample preparation.
- The sections have been corrected
Section 3.5. Reducing sugars: please add details. Also, the results of this section were not reported.
- The method has been described. The contents of reducing sugar were for the characterization of final wines, and are commented in new lines 190-195.
Line 388: what kind of acid was used for hydrolysis.
- The acid has been added (HCl)
Section 3.6.4. Ethanol content of wines: please add the results of this section.
- The content of ethanol was for the characterization of final wines, and are commented in new lines 190-195.
Round 2
Reviewer 2 Report
This version looks much better and the statistical analysis makes the results more persuasive.
Author Response
Thank you very much for your comments
Reviewer 3 Report
I would like to thank the authors for their work and corrections.
I only have minor remarks :
Abstract: You did not answer my previous comment: "please add the techniques used (DPPH ...). Some language corrections need to be made"
For statistics: Did you not analyze the results of juice and wine?
Line 56: what do you mean by "Authors as Prior et al. [4]"? If it's their work, you should say directly: "Prior et al. [4]".
Table 1 : Mv-3-glc: Malvidin-3-glucoside is repeated two times in the table foot (abbreviation). Also, please add in the table description (foot) the meaning of a, b, and c (statistics). Same for other tables or figures containing statistics.
Lines 75-77: It is still not clear what the authors mean by this idea. You have the five main anthocyanins also in the other families (glucoside and arabinoside). If you mean in the skin, you have them also in berries. Please make it more clear for readers.
Table 3: please edit the table form
Author Response
I would like to thank the authors for their work and corrections.
- Thank you very much for your comments
I only have minor remarks:
Abstract: You did not answer my previous comment: "please add the techniques used (DPPH ...). Some language corrections need to be made"
- We actually forgot to answer this suggestion. The techniques used in the summary have already been added.
For statistics: Did you not analyze the results of juice and wine?
- Statistical analysis has also been performed on the juice and wines, and has been included in this version.
Line 56: what do you mean by "Authors as Prior et al. [4]"? If it's their work, you should say directly: "Prior et al. [4]".
- The sentence has been corrected
Table 1 : Mv-3-glc: Malvidin-3-glucoside is repeated two times in the table foot (abbreviation). Also, please add in the table description (foot) the meaning of a, b, and c (statistics). Same for other tables or figures containing statistics.
- The repetition has been eliminated. The meaning of superscripts a, b and c has been included in all tables and figures.
Lines 75-77: It is still not clear what the authors mean by this idea. You have the five main anthocyanins also in the other families (glucoside and arabinoside). If you mean in the skin, you have them also in berries. Please make it more clear for readers.
- The peonidine derivative does not appear in either the glucoside or the arabinoside family. In these two families there are only 4 anthocyanins. If the reviewer deems it necessary, this sentence may be deleted.
Table 3: please edit the table form
- Table 3 has been edited